# Estimating the impact of missed colorectal cancer diagnoses on life expectancy in Minamisoma City following the 2011 triple disaster

Hiroaki Saito[1,2,*], Michio Murakami[3¤], Akihiko Ozaki[4,5], Yoshitaka Nishikawa[2,6,7], Toyoaki Sawano[5,8], Yuki Shimada[5,9], Sho Fujioka[2], Tianchen Zhao[1], Tomoyoshi Oikawa[9], Yukio Kanazawa[10], Masaharu Tsubokura[1,5]

1 Department of Radiation Health Management, Fukushima Medical University School of Medicine, Fukushima, Fukushima, Japan, 2 Department of Internal Medicine, Soma Central Hospital, Soma, Fukushima, Japan, 3 Department of Health Risk Communication, Fukushima Medical University School of Medicine, Fukushima, Fukushima, Japan, 4 Breast and Thyroid Center, Jyoban Hospital of Tokiwa Foundation, Iwaki, Fukushima, Japan, 5 Research Center for Community Health, Minamisoma Municipal General Hospital, Minamisoma, Fukushima, Japan, 6 Department of Health Informatics, Kyoto University School of Public Health, Kyoto, Japan, 7 Takemi Program in International Health, Harvard T. H. Chan School of Public Health, Boston, Massachusetts, United States of America, 8 Department of Surgery, Jyoban Hospital of Tokiwa Foundation, Iwaki, Fukushima, Japan, 9 Department of Neurosurgery, Minamisoma Municipal General Hospital, Minamisoma City, Fukushima, Japan, 10 Department of Gastroenterology, Minamisoma Municipal General Hospital, Minamisoma, Fukushima, Japan

¤ Current address: Center for Infectious Disease Education and Research (CiDER), The University of Osaka, Suita, Osaka, Japan
* h.saito0515@gmail.com

## Abstract

### Background

After the 2011 Great East Japan Earthquake, participation in colorectal cancer (CRC) screening significantly decreased in Minamisoma City, Fukushima Prefecture. However, the long-term health effects of this decline in screening participation have not been quantified. This study aims to construct a model to evaluate the impact of post-disaster decreases in CRC screening participation on population health.

### Methods

We utilized the population and CRC screening data targeting 40–74 years-old residents in Minamisoma City. We compared the actual screening participation in 2011 with projected participation rates based on pre-disaster levels to estimate the number of residents who missed screening due to the disaster. Based on national CRC screening performance data and stage-specific survival rates in Japan, we estimated the number of missed CRC cases and modeled the additional the loss of life expectancy (LLE) due to CRC resulting from a one-year delay in diagnosis.

**Data availability statement:** All relevant data are within the paper and its Supporting Information files.

**Funding:** This work was supported by the Research Project on the Health Effects of Radiation, organized by the Ministry of the Environment, Japan, and JSPS KAKENHI (grant number JP20H04354). The funder provided support in the form of personal fees for authors HS, AO, but did not have any additional role in the study design, data collection and analysis, decision to publish, or preparation of the manuscript. The specific roles of these authors are articulated in the 'author contributions' section. There was no additional external funding received for this study.

**Competing interests:** A.O. received personal fees from MNES, Inc. outside of the submitted work. H.S. received an honorarium from TAIHO Pharmaceutical Co., Ltd., outside of the submitted work. None of the authors have any competing interests in this article. This does not alter our adherence to PLOS ONE policies on sharing data and materials.

## Results

The estimated number of colorectal cancer cases that might have been missed due to decreased screening participation was 1.794 (95% uncertainty interval: 1.597 to 1.994) for men and 1.203 (0.931 to 1.491) for women. The missed detection opportunities estimated result in 0.428 (0.282 to 0.582) person-years [2.684 (1.793 to 3.604) years per 10,000 persons] and 0.229 (0.103 to 0.372) person-years [0.993 (0.450 to 1.608) years per 10,000 persons] of additional LLE for men and women, respectively. The estimated cost per life-year saved was $1.12 \times 10^6$ ($0.81 \times 10^6$ *to* $1.62 \times 10^6$ )yen for men and $3.65 \times 10^6$ ($2.02 \times 10^6$ *to* $7.19 \times 10^6$ ) yen for women, respectively.

## Conclusions

The calculated additional LLE due to missed CRC screening was relatively small but suggests preventive health services should be considered in disaster response planning. These findings provide a quantitative framework for evaluating health impacts of service disruptions.

---

## Introduction

Public health mass screening in asymptomatic populations endeavors to identify diseases in the early stages. These screenings encompass a broad spectrum of infectious and chronic diseases, with a particular focus on cancer, to facilitate early detection and treatment. The efficacy of such screening programs is often evaluated by several key indicators, including a reduction in cancer-related deaths, overall mortality rates, and disease prevalence within the population. Loss of life expectancy (LLE) is a reduction in life expectancy caused by a disease. It is one of the most useful indicators of the disease burden for individuals and populations because it focuses on the impact on remaining life expectancy rather than on measures evaluating specific periods, such as 5-year survival rates [1]. Moreover, it allows for comparison with other public health issues, prioritization, and calculation of cost-effectiveness [2]. Cancer screening is appropriate for cancer types with large LLEs in the population for which interventions are likely to improve LLE.

Colorectal cancer (CRC) is a significant global health concern, with screening being an effective measure to decrease CRC-related mortality [3]. In many countries, CRC is the leading cause of death [4]. For this reason, many countries and municipalities have introduced organized CRC screening [5]. Several indicators of CRC screening performance have been identified, including participation rates, colonoscopy uptake rates, and accuracy. Maintaining these performance indicators is an important challenge for countries and their local governments.

The need to address the difficulties in maintaining cancer screening and assessing the impact of such difficulties has become particularly important in recent years. The COVID-19 pandemic caused interruptions in cancer screening in many countries and municipalities [6,7]. Screening services may also be interrupted during major economic

crises [8,9]. Interruptions and poor performance of screening services in the medium-to-long term are expected to reduce the effectiveness of interventions in the population. However, few studies have examined the impact of such crisis-induced interruptions in screening on the health of the population, especially because of the diseases being screened.

In this context, it would be useful to investigate the impact of the Great East Japan Earthquake in 2011 on CRC screening and the target population in Minamisoma City, Fukushima Prefecture. Many residents of Minamisoma were forced to evacuate for an extended time due to the earthquake, tsunami, and subsequent nuclear accidents [10]. In this region, it has been proven that the risks of diseases, such as diabetes, which have been exacerbated by changes in the environment, such as long-term evacuations, significantly exceed the almost negligible risk of radioactive contamination [11]. Therefore, it is recognized that secondary health measures must be considered during disasters and nuclear accidents that necessitate such extensive, long-term evacuations. On the other hand, there is a scarcity of research on the impact of this disaster on local residents in terms of cancer screening and subsequent treatment, which are crucial preventive efforts. In fact, after the earthquake, even though the system for providing CRC screening was maintained, the screening rates dropped from 11.7% to 3.4% of the eligible population in Minamisoma City [12]. It is therefore hypothesized that individuals who might have had CRC detected earlier under pre-disaster conditions experienced delays in diagnosis due to decreased screening participation. Analyzing the risk to the health of the population of such a decrease in CRC screening participation is an important task in the prioritization of interventions and policy decisions on the health status of the population and in disaster planning in the same itself. This study addresses CRC screening, a practice annually recommended for both men and women across all municipalities in Japan and recognized for its established effectiveness. Using CRC screening as a one of cancer screening initiative, it models the city of Minamisoma—a community that faced diverse health impacts due to the earthquake, tsunami, and the Fukushima Daiichi nuclear power plant accident following the Great East Japan Earthquake—as a basis to simulate the effects of screening participation rates. Specifically, the study aims to estimate the number of CRC cases that might have been missed due to the disaster and to quantify the impact of these missed detections on population health. By utilizing changes in LLE due to CRC as an indicator, this research quantitatively evaluates the consequences of screening in comparison to other health metrics in a post-disaster context.

## Methods

This study aimed to estimate the number of CRC cases overlooked during the 2011 CRC screening program in Minamisoma City, the stage distribution of these missed cases, and the consequent LLE due to non-detection. To achieve this, we used historical data on CRC screening participation in Minamisoma City, comprehensive CRC screening data from Japan, and lifetable data. We focused on individuals aged 40–74 years who were eligible for municipal CRC screening in Minamisoma in 2011. The modalities and target population for CRC screening in this area have been documented in the previous study.[11] Colorectal cancer screening data of Minamisoma is compiled for those who participated in the annually recommended faecal immunochemical test (FIT). Colonoscopy is recommended if FIT is positive. Our analysis evaluated additional LLE cases resulting from undetected and consequently advanced CRC cases. We based our calculations on the assumption of a one-year delay in cancer detection, which is considered to have minimal impact. The additional LLE resulting from delayed CRC screening was calculated as the difference between the life expectancy of a patient who would have been diagnosed earlier if screening had continued as usual and that of a patient whose diagnosis was delayed by 1 year due to non-screening.

### Data and assumptions made for estimation

There are differences in the prevalence and stage distribution of CRC detected through screening and via symptoms. Consequently, our analysis specifically focused on CRC cases detected through screening. We hypothesized that the participants at the study site, which diminished due to the disaster, would exhibit a rate of screening-detected CRC and stage distribution similar to the national average in Japan. Our data on CRC detected via the fecal occult blood test were

derived from the 2009 fiscal year outcomes published by the Japanese Society for Gastrointestinal Cancer Screening [13]. This dataset provides insights into the CRC detection rate by the fecal occult blood method across different age groups and the stage distribution of CRC, as reported by 583 screening organizations nationwide. Furthermore, we referred to the data compiled by the National Council of Cancer Centers for the survival rates associated with CRC detected through screening [14,15]. Utilizing data on CRC diagnosed in 2007, we employed one-to-ten-year relative survival rates by age, sex, and CRC stage (stages I to IV) for individuals aged 40 years and above (S1 and S2 Tables). To estimate the potential number of CRC screening participants in 2011, when the disaster did not occur, we multiplied the eligible population for that year by the average participation rate from 2009 to 2010.

We accessed these data on 28/05/2022.

## Methods for calculating loss of life expectancy by CRC

**Life expectancy without additional risk.** The calculation method for life expectancy and life expectancy at risk is based on past research [11,16,17]. Life expectancy $e(x)$ at age $x$ years was calculated separately for males and females and was defined as:

$$T(x) = \int_x^\infty L(t)dt$$

$$e(x) = \frac{T(x)}{L(x)}$$

Where $L(x)$ is the number living at the start of an interval at age $x$ (100,000 live births) (Fig 1). $T(x)$ calculates the total person-years lived by the population from age $x$ to the end of life, essentially representing the integrated survival population beyond age $x$. We calculated these using the survival probabilities for Japanese males and females [18].

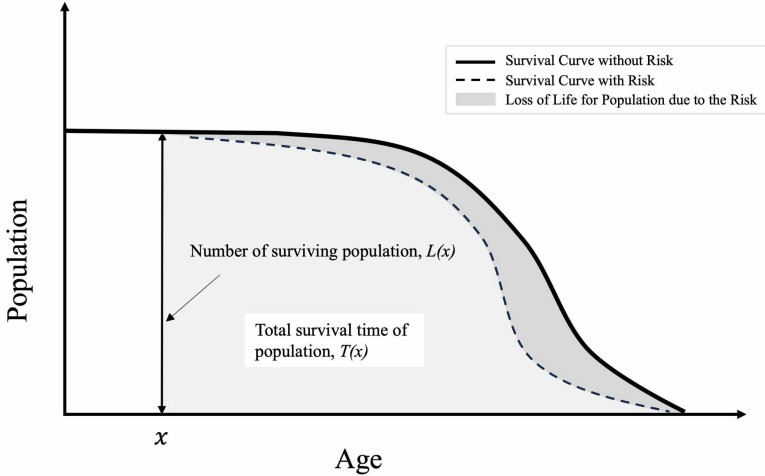

**Fig 1. Conceptual diagram of survival curve and loss of life expectancy.** The solid line represents the function $Y = L(x)$, which represents the number of individuals alive at age $x$ without the risk. The shaded area in light gray represents $T(x)$, which calculates the total person-years lived by the population from age $x$ to the end of life, essentially representing the integrated survival population beyond age $x$. The concept of lost life expectancy is illustrated through the area between the modified survival function $L'(x)$ (represented as a dashed line to account for risk factors) and the original survival function $L(x)$, indicating the difference in the integrated value of the population's survival, thus highlighting the impact of specific risks on overall life expectancy.

**Life expectancy with an additional risk of CRC.** The current analysis aimed to estimate the minimum additional LLE under the most conservative scenario that assessed the risk of CRC due to decreased screening. An increased risk of CRC mortality was assumed to occur in the first 10 years after detection. Life expectancy was calculated by multiplying the mortality rate at each age by the relative mortality rate at each stage of CRC, which was considered excess mortality due to screening-detected CRC. The calculation of the LLE due to CRC was defined as follows: Age at which cancer screening was performed, defined as "$a$"; if cancer screening was performed without delay: $a = 45, 55, 65, 72$ *years old*, and if cancer screening was delayed for 1 year: $a = 46, 56, 66, 73$ *years old*. We defined the following terms: $M(x, D)$ = mortality rate at age $x$ in CRC of stage $D$ (I, II, III, IV) in the patient population; $ND(x)$ = number of deaths at age $x$ at no additional risk; $NS(x)$ = number of survivors at age $x$ at no additional risk; and $RM(x, D)$ = relative mortality by the age $x$ and stage $D$. $M(x, D)$ was calculated as follows:

$$M(x, D) = \frac{(ND(x) + NS(x) \times RM(x, D))}{NS(x)}$$

Similarly, we defined the following terms: $l'(x, D)$ = number of those expected to live to reach age $x$ with an additional risk of stage $D$ CRC; $NSC(x, D)$ = number of survivors at age $x$ with an additional risk of stage $D$ CRC; and $NDC(x, D)$ = number of deaths at age $x$ with an additional risk of stage $D$ CRC. $l'(x, D)$ and $DC(x, D)$ were calculated as follows:

$$l'(x, D) = NSC(x, D) = NSC(x-1, D) - NDC(x-1, D)$$

$$NDC(x, D) = NSC(x, D) \times M(x, D)$$

The relative mortality rates of CRC vary according to the age at diagnosis, sex, and CRC stage. From $l'(x)$ and $L'(x, D)$, the number living at the start of an interval at age $x$ (of 100,000 born alive) with an additional risk of stage 'D' CRC was calculated using the survival probabilities for Japanese men and women:

$$L'(x, D) = \frac{11}{720} \times l'(x-2, D) - \frac{37}{360} \times l'(x-1, D) + \frac{19}{30} \times l'(x, D) + \frac{173}{360} \times l'(x+1, D) - \frac{19}{720} \times l'(x+2, D)$$

Life expectancy $e'(x, D)$ at age $x$ with an additional risk of stage 'D' CRC was defined as:

$$T'(x, D) = \int_{x}^{\infty} L'(t, D) dt$$

$$e'(x, D) = \frac{T'(x, D)}{L'(x, D)}$$

*LLE* $(x, D)$ for a person whose cancer was detected by cancer screening at age $x$ was defined as:

$$LLE(x, D) = e'(x, D) - e(x, D)$$

**Estimation of excess CRC deaths corresponding to a 1-year delay in diagnosis due to decreased screening**

Suppose $A(i)$ is the actual number of CRC screening participants in 2011 for a given age group I ($i = 40s, 50s, 60s, 70s[70 - 74 \ years]$). The hypothetical number of participants who underwent CRC screening in 2011 if

there were no disasters for age group '*i*,' *B*(*i*) can be calculated using the actual 2009 and 2010 CRC screening participation rates [$R_{2009}(i)$, $R_{2010}(i)$ and the 2011 eligible population $P_{2011}(i)$] as follows:

$$B(i) = P_{2011}(i) \times \frac{R_{2009}(i) + R_{2010}(i)}{2}$$

Thus, the number of CRC screening participants *P*(*i*) estimated to have decreased after the disaster can be expressed as:

$$P(i) = B(i) - A(i)$$

The proportion of CRC diagnosed among those who underwent CRC screening (fecal occult blood method) in Japan in a given age group (*i*) was defined as *Ci*, and the proportion of CRC detected by CRC screening in Japan in a given age group (*i*) included in stage 'D' was defined as *F*(*D*, *i*) (*D* = I, II, III, IV). The number of stage D patients who would have been detected in Minamisoma in a given age group (*i*) among the post-disaster decrease in CRC screening participants *R*(*D*, *i*) was calculated as follows:

$$R(D, \ i) = Pi \times Ci \times F(D, i)$$

The total number of CRCs detected by screening between the ages of 40 and 74 years *M*(*D*) was calculated as follows:

$$M(D) = R(D, i = 40s) + R(D, i = 50s) + R(D, i = 60s) + R(D, i = 70s)$$

LLE for age group (*i*), with stage*D* CRC if it had been discovered in 2011 (person-years) (*LLEC*(*D*, *i*)), was calculated as follows:

$$LLEC(D, i) = R(D, i) \times LLE(D, i)$$

***Estimated shift of CRC stage in case of reduced screening.*** According to a previous report, approximately 10% of CRCs in the early stages (stages I and II) progressed to advanced stages (stages III and IV) when the diagnosis was delayed for more than 1 year [7]. In the current simulation, we estimated the additional LLE that would occur under the assumption that 10% of CRCs included in the reduced number of participants in 2011 would move from an early stage (stages I, II) to an advanced stage (stages III and IV) in 1 year. A scenario for underestimating LLE was assumed, in which only 10% of individuals in stages I and II would move from stage II to III.
The number of patients with stage D CRC (*R'*(*D*, *i*)) after the stage shift was calculated as follows:

$$R'(I, \ i) = \ R(I, \ i)$$

$$R'(II, \ i) = \ R(II, \ i) - 0.1 \times \{R(I, \ i) + \ R(II, \ i)\}$$

$$R'(III, \ i) = \ R(III, \ i) + \ 0.1 \times \{R(I, \ i) + \ R(II, \ i)\}$$

$$R'(IV, \ i) = \ R(IV, \ i)$$

The change in LLE that would have been detected in 2011 for a given age group (*i*) as a whole, which increased with the stage transition of CRC, *Y*(*i*), was calculated as follows:

$$Y(i = 40s) = 0.1 \times \{R(I, 40s) + R(II, 40s)\} \times (LLE(III, x = 46)-- LLE(II, x = 45))$$

$$Y(i = 50s) = 0.1 \times \{R(I, 50s) + R(II, 50s)\} \times (LLE(III, x = 56)-- LLE(II, x = 55))$$

$$Y(i = 60s) = 0.1 \times \{R(I, 60s) + R(II, 60s)\} \times (LLE(III, x = 66)-- LLE(II, x = 65))$$

$$Y(i = 70s) = 0.1 \times \{R(I, 70s) + R(II, 70s)\} \times (LLE(III, x = 72)-- LLE(II, x = 73))$$

LLE due to delayed cancer screening per person (LLEP) was calculated as follows:

$$LLEP(i) = Y(i) \div B(i)$$

**Estimation of cost per life-year saved.** CPLYS is the cost-effectiveness indicator that measures the LLE spared for an intervention [11]. Minamisoma City requires each person to pay 400 yen (approximately 3.15 USD) for a fecal occult blood test. To calculate the CPLYS, this amount was calculated as the per capita cost of implementing CRC screening. Generally, it is calculated as the cost required to extend life expectancy. However, in this study, it was back-calculated from the LLE due to missed CRC screening and the cost of the screening not performed (CSR).

$$CPLYS = CSR \div LLEP$$

### Uncertainty estimation

To estimate uncertainty, Monte Carlo simulation was performed using Crystal Ball software (Oracle, Redwood City, CA, USA). For screening participation rates in Minamisoma City and national colorectal cancer detection rates, we calculated standard errors and defined probability distributions. For 10-year relative mortality rates of colorectal cancer, we utilized the reported 95% confidence intervals (S1 and S2 Tables). The simulation was conducted with 10,000 iterations for each age group and sex category. For each iteration, the model randomly sampled values from the defined probability distributions for all input parameters and calculated the resulting outcomes (number of missed CRC cases, stage distributions, and associated LLEs). The 95% uncertainty intervals (UIs) reported throughout our results represent the 2.5th and 97.5th percentiles of the output distributions from these 10,000 iterations. For key outputs such as the number of missed CRC diagnoses, the additional LLE due to stage progression, and the cost per life-year saved (CPLYS), we reported both the point estimates (based on the mean of the simulation results) and the 95% UIs.

### Ethical approval

The study was approved by the ethics committees of the Minamisoma Municipal General Hospital and Fukushima Medical University (approval nos. 2–20 and 3065, respectively). The need for informed consent was waived by the ethics committees of the Minamisoma Municipal General Hospital and Fukushima Medical University, owing to the retrospective nature of the study. All the experiments were performed in accordance with the Declaration of Helsinki.

### Results

### Loss of life expectancy from CRC by age and stage of cancer based on the Japanese national database

Based on the CRC screening results in Japan, LLE caused by CRC in patients in their 40s to 70s was calculated for each age group and sex. We summarized LLE in cases where the stage of CRC advanced within a year (Table 1).

**Table 1. Loss of life expectancy by age group and stage of colorectal cancer and change if detected after one year.**

| Age group | Sex | Colorectal cancer stage | LLE at baseline | LLE one year later |
|---|---|---|---|---|
| 40s | Male | Stage I | 2.086 (0.938-3.177) | 2.012 (0.921-3.064) |
| | | Stage II | 21.222 (16.614-24.722) | 20.666 (16.136-23.949) |
| | | Stage III | 25.838 (23.124-27.997) | 25.23 (22.321-27.125) |
| | | Stage IV | 33.997 (33.664-34.303) | 33.598 (32.663-33.305) |
| | | | | |
| | Female | Stage I | 15.081 (10.999-18.696) | 14.652 (10.708-18.170) |
| | | Stage II | 21.852 (16.949-25.847) | 24.995 (16.600-25.148) |
| | | Stage III | 37.862 (36.827-38.636) | 30.329 (35.897-37.648) |
| | | Stage IV | 40.317 (39.850-40.716) | 39.842 (38.845-39.706) |
| | | | | |
| 50s | Male | Stage I | 4.957 (3.312-6.459) | 4.824 (3.304-6.270) |
| | | Stage II | 9.585 (7.174-11.718) | 9.169 (6.844-11.185) |
| | | Stage III | 18.128 (16.700-19.383) | 17.352 (15.958-18.566) |
| | | Stage IV | 25.146 (24.958-25.326) | 24.153 (23.967-24.328) |
| | | | | |
| | Female | Stage I | 2.205 (1.449-2.946) | 2.119 (1.377-2.846) |
| | | Stage II | 16.709 (13.701-19.297) | 16.119 (13.135-18.661) |
| | | Stage III | 24.857 (23.627-25.891) | 23.981 (22.781-24.993) |
| | | Stage IV | 30.962 (30.679-31.218) | 29.962 (29.688-30.213) |
| | | | | |
| 60s | Male | Stage I | 3.570 (2.516-4.590) | 3.261 (2.260-4.205) |
| | | Stage II | 6.071 (4.597-7.396) | 5.658 (4.270-6.912) |
| | | Stage III | 10.867 (9.863-11.758) | 10.163 (9.225-10.991) |
| | | Stage IV | 16.999 (16.84-17.152) | 16.023 (15.866-16.171) |
| | | | | |
| | Female | Stage I | 3.806 (2.542-5.015) | 3.602 (2.407-4.750) |
| | | Stage II | 13.210 (11.333-14.815) | 12.559 (10.879-14.008) |
| | | Stage III | 14.877 (13.582-16.003) | 14.129 (12.897-15.182) |
| | | Stage IV | 22.077 (21.889-22.252) | 21.084 (20.898-21.260) |
| | | | | |
| 70s | Male | Stage I* | 0 | 0 |
| | | Stage II | 4.229 (3.176-5.177) | 3.759 (2.783-4.634) |
| | | Stage III | 5.879 (4.961-6.662) | 5.306 (4.455-6.021) |
| | | Stage IV | 11.959 (11.788-12.120) | 11.004 (10.558-11.418) |
| | | | | |
| | Female | Stage I* | 0 | 0 |
| | | Stage II | 9.212 (7.722-10.447) | 8.441 (7.088-9.627) |
| | | Stage III | 11.871 (11.065-12.572) | 11.041 (10.288-11.700) |
| | | Stage IV | 16.319 (16.097-16.525) | 15.332 (14.885-15.753) |

Values represent point estimates and 95% uncertainty intervals.

LLE: Loss of life expectancy (years).

LLE at baseline = LLE by colorectal cancer stage, age, and sex at diagnosis.

LLE one year later = LLE by colorectal cancer stage, age, and sex when the disease was diagnosed one year later.

*The relative survival rate for stage I colorectal cancer was 1.00, with an estimate of loss of life expectancy zero.

## Estimated simulation in Minamisoma City

In 2011, 426 men and 658 women aged 40–74 years underwent CRC screening. Applying this to the national CRC prevalence rate for CRC screening (S3 Table), the number of CRC cases detected in 2011 was expected to be 0.708 (95% Uncertainty interval [UI]: 0.667 to 0.749) for males and 0.488 (0.377 to 0.600) for females (Table 2). If the same average participation rates in 2009 and 2010 were applied in 2011, the number of male and female participants would have been 1593 (1491–1694) and 2306 (2188–2427), respectively. Furthermore, the number of detected CRCs was 2.502 (2.282 to 2.728) and 1.691 (1.313 to 2.087) in males and females, respectively. Therefore, it was estimated that 1.794 (1.597 to 1.994) males and 1.203 (0.931 to 1.491) females missed the diagnosis due to the decrease in the number of CRC screening participants after the disaster.

Table 3 shows the stage distribution of CRCs presumed to have been missed in 2011. Assuming that these CRCs were diagnosed in 2011, the estimated total LLE by CRC was 9.232 (7.988 to 10.557) years for males and 10.326 (7.671 to 13.130) years for females (Table 4). For men in their 40s, 50s, 60s, and 70s, the values were 0.358 (0.212 to 0.539), 1.366 (0.958 to 1.837), 5.826 (4.750 to 7.022), and 1.682 (1.314 to 2.08) years, respectively; for women, the LLE by CRC was 1.052 (0.178 to 2.030), 2.125 (0.690 to 3.700), 5.125 (3.278 to 7.104), and 2.025 (1.389 to 2.749) years, respectively (S1 Fig).

The lowest impact estimate for the progression of CRC in 1 year for those who missed the opportunity to be detected (with only stage II to III shift) was a total increase in LLE of 0.428 (0.282 to 0.582) person-years for men and 0.229 (0.103 to 0.372) for women (Table 5). Men in their 40s, 50s, 60s, and 70s had LLEs of 0.007 (−0.001 to 0.018), 0.072 (0.043 to 0.109), 0.293 (0.172 to 0.428), and 0.056 (−0.009 to 0.125) person-years, while females had LLEs of 0.047 (−0.007 to 0.096), 0.086 (−0.025 to 0.167), 0.04 (−0.047 to 0.139), and 0.056 (−0.011 to 0.113) person-years, respectively. For the population expected to be screened, men and women had a total increase in LLE of 2.684 (1.793 to 3.604) and 0.993 (0.450 to 1.608) years per 10,000 persons, respectively. Men in their 40s, 50s, 60s, and 70s experienced LLE increases of 0.573 (−0.095 to 1.417), 2.865 (1.795 to 4.120), 3.696 (2.203 to 5.277), and 1.298 (−0.217 to 2.868) years per 10,000 persons, respectively; females experienced LLE increases of 1.878 (0.297 to 3.765), 1.638 (0.480 to 3.149), 0.373 (−0.439 to 1.284), and 1.217 (0.245 to 2.412) years per 10,000 persons, respectively.

The total cost spared due to the decrease in the number of people screened was estimated to be 464,418 (423,926–505187) yen for males and 749,505 (701,587–796,835) yen for females. The estimated cost per life-year saved (CPLYS) was $1{,}120 \times 10^6$ ($0.808 \times 10^6$, $1.620 \times 10^6$) yen for men and $3.650 \times 10^6$ (95%CI: $2.018 \times 10^6$, $7.188 \times 10^6$) yen for women (Table 5).

**Table 2. Estimated detected and non-detected colorectal cancer rates and their clinical stages based on colorectal cancer screening in Minamisoma City in 2011.**

| | Estimate of detected colorectal cancer | | | | Estimate of non-detected cancer (B-A) | |
| --- | --- | --- | --- | --- | --- | --- |
| | Based on actual number of participants (A)* | | Based on the number of participants, same as before the earthquake (B)** | | | |
| Stage | Male | Female | Male | Female | Male | Female |
| I | 0.376 (0.351-0.400) | 0.257 (0.199-0.316) | 1.327 (1.204-1.454) | 0.890 (0.691-1.099) | 0.951 (0.843-1.062) | 0.633 (0.490-0.785) |
| II | 0.155 (0.140-0.170) | 0.106 (0.081-0.132) | 0.546 (0.485-0.613) | 0.366 (0.281-0.457) | 0.391 (0.341-0.446) | 0.260 (0.199-0.326) |
| III | 0.159 (0.145-0.175) | 0.112 (0.085-0.140) | 0.563 (0.500-0.631) | 0.389 (0.296-0.486) | 0.404 (0.353-0.460) | 0.277 (0.210-0.348) |
| IV | 0.018 (0.013-0.023) | 0.013 (0.009-0.018) | 0.065 (0.048-0.083) | 0.046 (0.031-0.062) | 0.047 (0.034-0.060) | 0.033 (0.022-0.045) |
| Total | 0.708 (0.667-0.749) | 0.488 (0.377-0.600) | 2.502 (2.282-2.728) | 1.691 (1.313-2.087) | 1.794 (1.597-1.994) | 1.203 (0.931-1.491) |

Values represent point estimates and 95% uncertainty intervals.

*Stage distribution of detected colorectal cancer estimated from the actual number of screening participants in 2011 (units: persons).

**Stage distribution of detected colorectal cancer based on estimates if screening had been performed in 2011 based on average screening participation rates in 2009 and 2010 (unit: persons).

**Table 3. Estimated distribution of colorectal cancer stage among patients assumed to have missed diagnosis in 2011 (unit: persons).**

| Stage | Male | | | | | Female | | | | |
|---|---|---|---|---|---|---|---|---|---|---|
| | Total† | 40–49 yr* | 50–59 yr | 60–69 yr | 70–74 yr | Total† | 40–49 yr | 50–59 yr | 60–69 yr | 70–74 yr |
| I | 0.951 (0.843-1.062) | 0.014 (0.008-0.021) | 0.065 (0.045-0.087) | 0.506 (0.426-0.593) | 0.367 (0.301-0.437) | 0.633 (0.490-0.785) | 0.023 (0.004-0.045) | 0.082 (0.026-0.143) | 0.308 (0.199-0.423) | 0.220 (0.151-0.295) |
| II | 0.391 (0.341-0.446) | 0.005 (0.002-0.008) | 0.028 (0.019-0.039) | 0.210 (0.171-0.251) | 0.149 (0.119-0.181) | 0.260 (0.199-0.326) | 0.008 (0.001-0.017) | 0.036 (0.011-0.064) | 0.127 (0.081-0.178) | 0.089 (0.061-0.121) |
| III | 0.404 (0.353-0.460) | 0.007 (0.004-0.011) | 0.038 (0.026-0.051) | 0.211 (0.173-0.254) | 0.148 (0.119-0.181) | 0.277 (0.210-0.348) | 0.011 (0.002-0.023) | 0.048 (0.015-0.084) | 0.129 (0.081-0.180) | 0.089 (0.061-0.121) |
| IV | 0.047 (0.034-0.060) | 0.002 (0-0.003) | 0.004 (0.001-0.007) | 0.026 (0.016-0.037) | 0.015 (0.009-0.022) | 0.033 (0.022-0.045) | 0.003 (0-0.006) | 0.005 (0.001-0.010) | 0.016 (0.008-0.025) | 0.009 (0.005-0.014) |
| Total | 1.794 (1.597-1.994) | 0.027 (0.017-0.039) | 0.134 (0.095-0.176) | 0.954 (0.809-1.111) | 0.679 (0.56-0.805) | 1.203 (0.931-1.491) | 0.045 (0.008-0.086) | 0.171 (0.055-0.295) | 0.580 (0.376-0.796) | 0.407 (0.282-0.543) |

Values represent point estimates and 95% uncertainty intervals.

*yr: years.

†The 'Total' column in the table represents the summation of loss of life expectancy (LLE) estimates across all age groups for the whole population.

**Table 4. Loss of life expectancy that would result if colorectal cancer missed in 2011 had been diagnosed in 2011 (unit: years).**

| Stage | Male | | | | | Female | | | | |
|---|---|---|---|---|---|---|---|---|---|---|
| | Total† | 40-49 yr* | 50-59 yr | 60-69 yr | 70-74 yr | Total† | 40-49 yr | 50-59 yr | 60-69 yr | 70-74 yr |
| I | 2.156 (1.553-2.794) | 0.029 (0.011-0.052) | 0.320 (0.190-0.476) | 1.807 (1.222-2.436) | 0 (0−0) | 1.701 (1.084-2.421) | 0.348 (0.056-0.707) | 0.181 (0.054-0.343) | 1.171 (0.654-1.795) | 0 (0−0) |
| II | 2.274 (1.840-2.740) | 0.101 (0.050-0.169) | 0.271 (0.166-0.401) | 1.274 (0.905-1.681) | 0.628 (0.438-0.837) | 3.278 (2.429-4.203) | 0.173 (0.027-0.370) | 0.602 (0.185-1.095) | 1.684 (1.055-2.401) | 0.82 (0.543-1.149) |
| III | 4.024 (3.465-4.641) | 0.174 (0.095-0.275) | 0.680 (0.462-0.936) | 2.297 (1.843-2.817) | 0.873 (0.663-1.111) | 4.588 (3.355-5.904) | 0.424 (0.070-0.857) | 1.192 (0.380-2.100) | 1.915 (1.207-2.693) | 1.057 (0.711-1.447) |
| IV | 0.778 (0.567-0.998) | 0.054 (0.017-0.101) | 0.095 (0.032-0.168) | 0.448 (0.273-0.636) | 0.181 (0.102-0.266) | 0.758 (0.504-1.056) | 0.106 (0.012-0.249) | 0.150 (0.032-0.321) | 0.354 (0.186-0.557) | 0.148 (0.078-0.234) |
| Total | 9.232 (7.988-10.557) | 0.358 (0.212-0.539) | 1.366 (0.958-1.837) | 5.826 (4.750-7.022) | 1.682 (1.314-2.08) | 10.326 (7.671-13.130) | 1.052 (0.178-2.030) | 2.125 (0.690-3.700) | 5.125 (3.278-7.104) | 2.025 (1.389-2.749) |

Values represent point estimates and 95% uncertainty intervals.

*yr: years.

†The column "Total" provides estimates of total loss of life expectancy for the entire population, which is equal to the sum of those for each age group.

## Discussion

In this study, we estimated for the first time the change in LLE due to CRC resulting from a decrease in screening after the 2011 Great East Japan Earthquake. Although the estimated number of undiagnosed CRCs due to decreased screening was small, this study allowed us to quantify the risk of decreased screening after a disaster. Quantifying the impact of this change in CRC screening as an indicator for prioritizing health issues in this region, which has experienced a major disaster and nuclear accident, is very significant [13,19].

The additional LLE estimated from the reduction in CRC screening participation after the disaster was relatively lower than the health risks previously reported in relation to the Great East Japan Earthquake, although the present estimation is based on the most conservative scenario. For example, the additional LLE due to additional diabetes increases during the first 10 years after the disaster was reported to be $8.0 \times 10^{-2}$ years for the 40–74-year-old population [11]. The additional colorectal-cancer-related LLE due to a decrease in the screening participation in a year revealed in the study were

**Table 5. Additional loss of life expectancy and cost per life-year saved resulting from changes in colorectal cancer stage due to a 1-year diagnostic delay.**

| | Male | | | | | Female | | | | |
|---|---|---|---|---|---|---|---|---|---|---|
| | Total | 40s | 50s | 60s | 70s | Total | 40s | 50s | 60s | 70s |
| Additional LLE* (person-years) | 0.428 (0.282 −0.582) | 0.007 (−0.001 - 0.018) | 0.072 (0.043 - 0.109) | 0.293 (0.172 −0.428) | 0.056 (−0.009 - 0.125) | 0.229 (0.103 - 0.372) | 0.047 (0.007 - 0.096) | 0.086 (0.025 - 0.167) | 0.04 (−0.047 −0.139) | 0.056 (0.011 - 0.113) |
| Additional LLE per 10000 person** (years) | 2.684 (1.793 −3.604) | 0.573 (−0.095 - 1.417) | 2.865 (1.795 −4.120) | 3.696 (2.203 −5.277) | 1.298 (−0.217 - 2.868) | 0.993 (0.450 - 1.608) | 1.878 (0.297 −3.765) | 1.638 (0.480 - 3.149) | 0.373 (−0.439 - 1.284) | 1.217 (0.245 −2.412) |
| CPLYS, yen per LY** (yen/years) | 1120460 (808463 - 1620350) | 6758751 (−38445979 - 57058168) | 1049268 (702554 - 1590756) | 807953 (539145 −1287906) | 2356563 (−11971921 - 17702062) | 3650460 (2018426 - 7188361) | 4054515 (1007700 - 12755663) | 2787015 (1099044 - 7244178) | 6897377 (−84623673 - 91884349) | 2661304 (1182015 - 10343411) |

Values represent point estimates and 95% uncertainty intervals.

*LLE: Loss of life expectancy (years).

** LLE divided by the number of expected screening participants.

***CPLYS: Cost per life-year saved.

$2.684 \times 10^{-4}$ for males and $0.993 \times 10^{-4}$ for females, respectively. Thus, the decrease in CRC screening in 1 year is about 1/300–1/800 of the risk of an increase in diabetes in the 10 years after the earthquake. Therefore, if the impact of the decline in screening participation only stops after 1 year in Minamisoma City, the overall health indicators of the population could be improved by focusing on measures such as diabetes control. However, as discussed below, as the CPLYS obtained in this study shows, from a cost-effectiveness perspective, the local government has a basis for recommending CRC screening because it is an effective intervention to improve LLE at an affordable cost. The desired intervention is a subject for future research, but it would be desirable to consider implementing a reintroduction of screening awareness and enhanced follow-up programmes to restore screening participation rates. In addition, there are many processes and limitations to the current risk calculation, and the actual impact of lower CRC screening uptake in the current study field may have occurred over an even longer period, ultimately necessitating a comprehensive assessment of such impacts. The strength of the current study is that it considered the risk of a decline in CRC screening in the year after the disaster, quantified it, and made it comparable to other indicators.

The application of LLE as a metric for assessing the effectiveness of CRC screening is notably distinctive. While excess deaths are commonly employed to gauge the increased burden of disease, these measures face limitations when comparing the impacts of conditions with varying temporal effects, such as radiation exposure, CRC, or chronic diseases. This is partly because it is challenging to compare excess deaths across these different contexts. In the aftermath of the disasters in Minamisoma City, it has been crucial to maintain or enhance the health status of evacuees over the long term. The validity of measuring LLE is supported by two perspectives: health-maximizing ageism [20] and fair-innings ageism [21]. These concepts assert that humans are entitled to a certain level of life expectancy and that the impact on their health should be assessed over their entire lifetime rather than at a single point in time. From this viewpoint, it is significant that LLE was employed in this study to conduct the evaluation. Given the variety of health impacts observed after the disaster, such as weight gain, diabetes, hypertension, dyslipidemia, and psychological stress [22–24], followed by increased susceptibility to cardiac, cerebrovascular, and cancer diseases, employing LLE and CPLYS as a common measure proves invaluable in identifying and prioritizing interventions for detailed study.

The CPLYS obtained in this study demonstrated the effectiveness of CRC screening. After the nuclear accident, residents experienced food shipment restrictions and decontamination as long-term health measures; the CPLYSs were estimated to be $5.6 \times 10^{7}$ and $2.4 \times 10^{8}$ yen, respectively, [11] which was more cost-consuming than CRC screening, as shown in the present study. The CPLYS for CRC screening calculated in this study is comparable to the reported

cost of general health checkups and conservative management for diabetes ($<7.4 \times 10^6$ yen per year) [11]. Generally, interventions for patients are more cost-effective than those for chemical pollution control or disease prevention strategies because they are more targeted [25]. In addition, the CPLYS measured in this study only considered the cost-effectiveness of CRC screening; we did not consider the costs associated with treating CRC. In Japan, CRC treatment, whether in the curative or non-curative stage, is relatively expensive [26,27]. Only screening expenses were included in this estimate, and expenditures associated with CRC treatment were excluded. This is because we expected that treatment would be required and that a one-year discovery delay would not affect medical expenditures. If greater stage advancement or the emergence of new CRCs occurs as a result of screening delays, the related increase in medical expenditure for treatment would need to be factored in to assess cost-effectiveness more accurately. Although many other public health issues have been identified, this study shows that municipalities should be proactive in increasing CRC screening rates.

According to previous studies, declining CRC screening can have a greater long-term impact. A scenario that assumed CRC screening suspension for 12 months due to a COVID-19 outbreak reported an additional LLE of $6.59 \times 10^{-3}$ to $2.94 \times 10^{-2}$ years per person, based on three CRC simulation models from Australia, Canada, and New Zealand [6]. This is a greater estimate of the impact of cessation of CRC screening on LLE than that of the current study. Our study may have underestimated LLE due to decreased CRC screening participation because the effect of screening on the detection of precancerous lesions was not examined. The best assumption that CRC will be identified 1 year later if missed was made. The reduction in CRC screening participation rates has been ongoing for more than 2 years since the disaster, [12] and the target population is greater than estimated; hence, the trend of missed CRCs may be considerably further along. Therefore, the actual impact of the decline in CRC screening participation rates after a disaster could be much higher. Additionally, LLE depends on the subsequent treatment outcomes of the disease. The results of this study could be used to simulate the decline in CRC screening participation rates in developed countries with comparable levels of medical care and similar population compositions to those in Japan.

Regarding additional LLE owing to lower screening participation rates, there were differences in age-specific trends between males and females. Male LLE per participant increased with age, while female LLE per participant was highest in those in their 40s, indicating a difference between sexes in the current study. In Japan, CRC is more prevalent in men than in women, and its incidence increases with age. Conversely, the value of LLE is higher in patients diagnosed at younger ages. CRC is less common at a younger age but sometimes more likely to be found at an advanced stage [28]. In Japan, there are few survival data on young-onset CRC, and accumulating more data is necessary for more accurate predictions. The increase of young-onset CRC patients is a global problem, requiring prevention and countermeasures [29,30]. In particular, non-participation in CRC screening after the 2011 disaster was more likely to occur in the working-age population aged 40–64 years than in those ≥65 years [12]. It is important to educate the younger generation about the possibility of CRC and to increase their participation rate.

This study has several limitations. First, in calculating LLE, we assumed that the proportion of CRC cases detected through the fecal occult blood method in the study population would match the national statistics for Japan. This assumption implies that if the population risk is equivalent, and the primary screening positive rate and secondary screening attendance rate are stable, the number of CRC diagnoses can be predicted for any population undergoing the screening. However, this may not hold true for other populations. Notably, Japan has a known low colonoscopy screening attendance rate, which could be higher in populations with better educational approaches to CRC screening. When considering the generalizability of this research method to other populations, it is crucial to evaluate whether the population's CRC risk and indicators for each screening process are comparable to this study. Adjusting these factors as needed may enhance the generalizability of this study. Second, this study does not account for the effect of identifying and excising colorectal polyps, which are precancerous CRC lesions. The impact of CRC screening encompasses both early CRC detection to suppress cancer progression and the prevention of CRC development by detecting and

removing precancerous lesions. Our estimations only include the effect of reduced life expectancy due to CRC detected through screening and do not consider the preventive effect of identifying and treating precancerous lesions. Consequently, the increase in LLE due to the decline in CRC screening participation may be underestimated. However, the time from advanced adenoma to CRC death is estimated to be 10–15 years [31], suggesting a negligible impact within our model's timeframe. To estimate the effect of LLE, including the treatment effect of precancerous lesions, a model that evaluates long-term impacts such as microsimulation models [32], needs to be constructed. Third, the estimations were based on CRC survival data from 2007, the most recent year for which data were available and closely aligned with the target year for the study. In this data, the relative 10-year survival rate for stage I CRC in the 70s was 1. These numbers may differ with advancements in treatment, changes in the healthcare environment, and the number of patients enrolled in the future. Furthermore, by using these national average relative survival data, we adapt the estimation that the study area is served by average medical care in Japan. It is possible that the level of medical care in the area may have been affected by a lack of medical resources after the disaster and that the results of this study may be an underestimate of the reality. Fourth, information regarding the progression of colorectal cancer stages due to a one-year delay in diagnosis is based on limited literature. The assumption that 10% of early-stage (I and II) colorectal cancers would progress to more advanced stages within one year may not fully capture the complex nature of cancer progression, which can vary significantly between individuals and cancer subtypes. Future research with more robust data on stage progression rates specific to the Japanese population would strengthen our estimation. Finally, the Minamisoma City population was < 70,000 at the time of the disaster, which is a relatively small target population. This was compounded by the fact that the CRC screening participation rate was low (in the 10% range) before the disaster, suggesting a potentially limited number of study participants. These factors introduced significant limitations during the verification process. However, to mitigate these limitations, we employed Monte Carlo analysis to enhance the validity of our research.

## Conclusions

Calculated LLE based on the decreased CRC screening participation rates in Minamisoma City during the first year following the disaster was relatively small. However, a low CPLYS calculated in the present study signifies that the measure is highly cost-effective, further underscoring the importance of recommending CRC screening. A more extended and comprehensive analysis is required to capture the entire impact of the disaster on CRC screening. Using LLE to prioritize cancer screenings in post-disaster regions, in comparison with the impacts of other diseases, is likely to be important. This approach can help allocate resources effectively, focusing on interventions that offer the greatest potential for extending life expectancy within the affected population.

## Supporting information

**S1 Table. Relative survival rates for men by age and stage of colorectal cancer.**
(DOCX)

**S2 Table. Relative survival rates for women by age and stage of colorectal cancer.**
(DOCX)

**S3 Table. Distribution of colorectal cancers detected by colorectal cancer mass screening in Japan by stage.**
(DOCX)

**S1 Fig. Additional loss of life expectancy resulting from changes in colorectal cancer stage due to a 1-year diagnostic delay per 10000 persons.**
(DOCX)

 

## Acknowledgments

We would like to thank Ms. Yuka Harada and Mr. Masatsugu Tanaki for managing the data for this study, as well as the medical staff of Minamisoma Municipal Hospital for their cooperation in the CRC screening program.

## Author contributions

**Conceptualization:** Hiroaki Saito, Michio Murakami.

**Formal analysis:** Hiroaki Saito.

**Funding acquisition:** Masaharu Tsubokura.

**Investigation:** Hiroaki Saito.

**Supervision:** Akihiko Ozaki, Masaharu Tsubokura.

**Writing – original draft:** Hiroaki Saito.

**Writing – review & editing:** Michio Murakami, Akihiko Ozaki, Yoshitaka Nishikawa, Toyoaki Sawano, Sho Fujioka, Tianchen Zhao, Tomoyoshi Oikawa, Yukio Kanazawa, Masaharu Tsubokura, Yuki Shimada.

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
