## [Decision Letter · Decision Letter 0]

23 Sep 2024

Dear Dr. Saito,

Thank you for submitting your manuscript to PLOS ONE. After careful consideration, we feel that it has merit but does not fully meet PLOS ONE’s publication criteria as it currently stands. Therefore, we invite you to submit a revised version of the manuscript that addresses the points raised during the review process.

We look forward to receiving your revised manuscript.

Kind regards,

Belgin Sever, Ph.D.

Academic Editor

PLOS ONE

**Journal Requirements:**

This work was supported by the Research Project on the Health Effects of Radiation, organized by the Ministry of the Environment, Japan, and JSPS KAKENHI (grant number JP20H04354). 

I have read the journal's policy and the authors of this manuscript have the following competing interests:A.O. received personal fees from MNES, Inc. outside of the submitted work. H.S. received an honorarium from TAIHO Pharmaceutical Co., Ltd., outside of the submitted work. None of the authors have any competing interests in this article.

We note that one or more of the authors are employed by a commercial company. 

4. In the online submission form, you indicated that your data is available only on request from a third party. Please note that your Data Availability Statement is currently missing the contact details for the third party, such as an email address or a link to where data requests can be made. Please update your statement with the missing information. 

Reviewers' comments:

Reviewer's Responses to Questions

**Comments to the Author**

1. Is the manuscript technically sound, and do the data support the conclusions?

Reviewer #1: Yes

2. Has the statistical analysis been performed appropriately and rigorously?

Reviewer #1: Yes

3. Have the authors made all data underlying the findings in their manuscript fully available?

Reviewer #1: Yes

4. Is the manuscript presented in an intelligible fashion and written in standard English?

Reviewer #1: Yes

**Reviewer #1:**  Your manuscript provides a valuable contribution to the field of disaster medicine and public health. The modeling approach and analysis are rigorous, and the policy implications of your findings are clear. To further strengthen your paper, consider addressing the points mentioned in the editor's recommendation, particularly expanding on the limitations of your assumptions and the potential impact of missed screenings on precancerous lesions.

I recommend accepting the manuscript with minor revisions, particularly focusing on the following areas:

1. Enhance the discussion of limitations: Expand on the limitations related to the assumptions made in the LLE calculations and the generalizability of the findings to other populations.

2. Clarify the impact of screening on precancerous lesions: Address the potential impact of missed CRC screenings on the detection and removal of precancerous lesions, which could further influence the study's conclusions.

3. Include a brief mention of limitations in the abstract: Add a sentence in the abstract that briefly acknowledges the study's limitations.

Overall, this is a well-executed study that has the potential to inform public health policies in disaster-affected regions. I look forward to seeing the final version of your manuscript.

**Do you want your identity to be public for this peer review?** For information about this choice, including consent withdrawal, please see our Privacy Policy

Reviewer #1: **Yes: ** NIKOLAOS GOUVAS

---

## [Author Response · Author response to Decision Letter 1]

12 Nov 2024

Response to the comments

Reviewers' comments:

Reviewer #1: Your manuscript provides a valuable contribution to the field of disaster medicine and public health. The modeling approach and analysis are rigorous, and the policy implications of your findings are clear. To further strengthen your paper, consider addressing the points mentioned in the editor's recommendation, particularly expanding on the limitations of your assumptions and the potential impact of missed screenings on precancerous lesions.

Response

We appreciate your comments. We have modified the manuscript according to the editor’s comments and reviewer’s comments.

Comments

I recommend accepting the manuscript with minor revisions, particularly focusing on the following areas:

1. Enhance the discussion of limitations: Expand on the limitations related to the assumptions made in the LLE calculations and the generalizability of the findings to other populations.

Response

Thank you for this helpful comment. We have added to the limitations section a discussion of our assumption that the proportion of colorectal cancers detected through screening in our study population would match national statistics. We acknowledge that this assumption may not hold true for other populations. To address the generalizability of our findings, we further elaborated on the need to consider key process indicators related to colorectal cancer screening, such as population risk, primary screening positive rate, and secondary screening attendance rate. When applying our methodology to other populations, these factors should be carefully evaluated and adjustments made as needed to ensure the validity of the results. We have added the following sentences.

Page35 Lines109-120

This study has several limitations. First, in calculating LLE, we assumed that the proportion of CRC cases detected through the fecal occult blood method in the study population would match the national statistics for Japan. This assumption implies that if the population risk is equivalent, and the primary screening positive rate and secondary screening attendance rate are stable, the number of CRC diagnoses can be predicted for any population undergoing the screening. However, this may not hold true for other populations. Notably, Japan has a known low colonoscopy screening attendance rate, which could be higher in populations with better educational approaches to CRC screening. When considering the generalizability of this research method to other populations, it is crucial to evaluate whether the population's CRC risk and indicators for each screening process are comparable to this study. Adjusting these factors as needed may enhance the generalizability of this study.

Comments

2. Clarify the impact of screening on precancerous lesions: Address the potential impact of missed CRC screenings on the detection and removal of precancerous lesions, which could further influence the study's conclusions.

Response

Thank you for your valuable comment. As you pointed out, one of the limitations of this study is that it does not include the therapeutic effect of precancerous lesions as a cancer prevention effect. We have added further details regarding this point to the limitations section. In this model, we calculated the effect on the LEE of detected colorectal cancer over a 10-year span. On the other hand, it is generally considered that it takes at least 10-15 years for an adenoma to lead to colorectal cancer death, so we believe it is acceptable not to include it in this model. Certainly, a model that measures the long-term effects of these treatments is needed, and we have indicated the need for further research.

Page35-6 Lines121-132

Second, this study does not account for the effect of identifying and excising colorectal polyps, which are precancerous CRC lesions. The impact of CRC screening encompasses both early CRC detection to suppress cancer progression and the prevention of CRC development by detecting and removing precancerous lesions. Our estimations only include the effect of reduced life expectancy due to CRC detected through screening and do not consider the preventive effect of identifying and treating precancerous lesions. Consequently, the increase in LLE due to the decline in CRC screening participation may be underestimated. However, the time from advanced adenoma to CRC death is estimated to be 10-15 years [31], suggesting a negligible impact within our model's timeframe. To estimate the effect of LLE, including the treatment effect of precancerous lesions, a model that evaluates long-term impacts such as microsimulation models [32], needs to be constructed.

Comments

3. Include a brief mention of limitations in the abstract: Add a sentence in the abstract that briefly acknowledges the study's limitations.

Response

We appreciate your comment. We have modified the abstract as follows.

Page 37 Lines 149-159

Conclusions: Although these data could be used to determine health policy goals following a catastrophic disaster, they should be interpreted with caution because they may underestimate the impact of CRC screening. This study has limitations, including the assumption of uniform national CRC screening and treatment performance and the exclusion of the preventive effect of polyp removal.

---

## [Decision Letter · Decision Letter 1]

12 Mar 2025

Dear Dr. Saito,

Thank you for submitting your manuscript to PLOS ONE. After careful consideration, we feel that it has merit but does not fully meet PLOS ONE’s publication criteria as it currently stands. Therefore, we invite you to submit a revised version of the manuscript that addresses the points raised during the review process.

We look forward to receiving your revised manuscript.

Kind regards,

Ahmed E. Abdel Moneim

Academic Editor

PLOS ONE

Journal Requirements:

Reviewers' comments:

Reviewer's Responses to Questions

**Comments to the Author**

Reviewer #2: (No Response)

Reviewer #3: All comments have been addressed

Reviewer #4: All comments have been addressed

2. Is the manuscript technically sound, and do the data support the conclusions?

Reviewer #2: Partly

Reviewer #3: Yes

Reviewer #4: No

3. Has the statistical analysis been performed appropriately and rigorously?

Reviewer #2: Yes

Reviewer #3: Yes

Reviewer #4: Yes

4. Have the authors made all data underlying the findings in their manuscript fully available?

Reviewer #2: No

Reviewer #3: Yes

Reviewer #4: No

5. Is the manuscript presented in an intelligible fashion and written in standard English?

Reviewer #2: Yes

Reviewer #3: Yes

Reviewer #4: No

Reviewer #2: Overall representation is good, the novelty, description and paper sections are managed well.

Only few things need to be considered to make it more valuable:

• Lack of Crystal Ball software Tool configuration

• Also include the limitations of Estimated shift of CRC stage in case of reduced screening.

• The visualization is not enough (recommendations: Bar Plotting OR Spider Plotting)

Reviewer #3: Very nice and methodologically sound and transparent paper on missed cancer cases in screening program due to a national disaster. I recommend it for publication.

Reviewer #4: I don’t like the abstract I will look review detail if you give go ahead ( no response)

Importsnt topic very much in my field. I was a silent academic interacting w the interested screeners

I don’t like the abstract

However ,Yes, I appreciate and very much sympathize w the authors cultural requirements

The abstract is 1.overloaded details re methods Results

2. This will be dangerously lost (misled)on reader uncertain sbout colonoscopy in their practices (expert opinion public heslth actor needs to go to text)

3.Please it needs lacks clear statements- tech jargon is not for abstract section

A)how many cancers were discovered at later staged updtsged (what stage 4,3,2 /10000 adults

B)which population is target for extra intervention (young or elderly define age cut

C)How many otherwise avoidable deaths !!

D)Is this enough evidence health loss , to support which/ sny salvage intervention ?

Whose (why do) criteria support a special intervention ?

E) did they adjust for real - world routine non compliance

If they are concerned that they underestimated the impsct this is a very concerning message ( this is a point for limitations not abstract )

Do They by extension dangerously downplay screening

( I don’t like that To msny doubt provokers come across my desk! History changing tech makes them wrong advice going forward

**Do you want your identity to be public for this peer review?** For information about this choice, including consent withdrawal, please see our Privacy Policy

Reviewer #2: **Yes: ** Muhammad Arshad

Reviewer #3: **Yes: ** Benjamin Benzon

Reviewer #4: No

---

## [Author Response · Author response to Decision Letter 2]

23 Apr 2025

Journal Requirements:

Reply

For the following cited papers, minor corrections to the terminology description in the abstract have been declared, but not retraceted. We therefore cited the article.

Satoh H, Ohira T, Hosoya M, Sakai A, Watanabe T, Ohtsuru A, et al. Evacuation after the Fukushima Daiichi Nuclear Power Plant Accident Is a Cause of Diabetes: Results from the Fukushima Health Management Survey. J Diabetes Res. 2015;2015:627390. Epub 2015/06/25. doi: 10.1155/2015/627390. PubMed PMID: 26106625; PubMed Central PMCID: PMCPMC4461763.

Comments

Reviewer #2: Overall representation is good, the novelty, description and paper sections are managed well.

Only few things need to be considered to make it more valuable:

• Lack of Crystal Ball software Tool configuration

Reply

Thank you for your comment. We have amended the description of the Monte Carlo simulation with Crystal Ball as follows.

Page 19 line 325-338

To estimate uncertainty, Monte Carlo simulation was performed using Crystal Ball software (Oracle, Redwood City, CA, USA). For screening participation rates in Minamisoma City and national colorectal cancer detection rates, we calculated standard errors and defined probability distributions. For 10-year relative mortality rates of colorectal cancer, we utilized the reported 95% confidence intervals (S1, S2 tables). The simulation was conducted with 10,000 iterations for each age group and sex category. For each iteration, the model randomly sampled values from the defined probability distributions for all input parameters and calculated the resulting outcomes (number of missed CRC cases, stage distributions, and associated LLEs). The 95% uncertainty intervals (UIs) reported throughout our results represent the 2.5th and 97.5th percentiles of the output distributions from these 10,000 iterations. For key outputs such as the number of missed CRC diagnoses, the additional LLE due to stage progression, and the cost per life-year saved (CPLYS), we reported both the point estimates (based on the mean of the simulation results) and the 95% UIs.

Comments

• Also include the limitations of Estimated shift of CRC stage in case of reduced screening.

Reply

Thank you for your suggestion. We have added this issue in the limitation as follows.

Page 37, line 143-150

Fourth, information regarding the progression of colorectal cancer stages due to a one-year delay in diagnosis is based on limited literature. The assumption that 10% of early-stage (I and II) colorectal cancers would progress to more advanced stages within one year may not fully capture the complex nature of cancer progression, which can vary significantly between individuals and cancer subtypes. Future research with more robust data on stage progression rates specific to the Japanese population would strengthen our estimation.

Comments

• The visualization is not enough (recommendations: Bar Plotting OR Spider Plotting)

Reply

In addition to the Table, additional loss of life expectancy resulting from changes in colorectal cancer stage due to a 1-year diagnostic delay has been added to the S1 figure.

Comments

Reviewer #3: Very nice and methodologically sound and transparent paper on missed cancer cases in screening program due to a national disaster. I recommend it for publication.

Reply

We appreciate your review and comments.

Comments

Reviewer #4: I don’t like the abstract I will look review detail if you give go ahead (no response)

Importsnt topic very much in my field. I was a silent academic interacting w the interested screeners

I don’t like the abstract

However ,Yes, I appreciate and very much sympathize w the authors cultural requirements

The abstract is 1.overloaded details re methods Results

Reply

We appreciate your comments. We have modified the abstract as follows.

Background: After the 2011 Great East Japan Earthquake, participation in colorectal cancer (CRC) screening significantly decreased in Minamisoma City, Fukushima Prefecture. However, the long-term health effects of this decline in screening participation have not been quantified. This study aims to construct a model to evaluate the impact of post-disaster decreases in CRC screening participation on population health.

Methods: We utilized the population and CRC screening data targeting 40-74 years-old residents in Minamisoma City. We compared the actual screening participation in 2011 with projected participation rates based on pre-disaster levels to estimate the number of residents who missed screening due to the disaster. Based on national CRC screening performance data and stage-specific survival rates in Japan, we estimated the number of missed CRC cases and modeled the additional the loss of life expectancy (LLE) due to CRC resulting from a one-year delay in diagnosis.

Results: The estimated number of colorectal cancer cases that might have been missed due to decreased screening participation was 1.794 (95% uncertainty interval: 1.597 to 1.994) for men and 1.203 (0.931 to 1.491) for women. The missed detection opportunities estimated result in 0.428 (0.282 to 0.582) person-years [2.684 (1.793 to 3.604) years per 10,000 persons] and 0.229 (0.103 to 0.372) person-years [0.993 (0.450 to 1.608) years per 10,000 persons] of additional LLE for men and women, respectively. The estimated cost per life-year saved was 1.12×10^6 (0.81×10^6 to 1.62×10^6 ) yen for men and 3.65×10^6 (2.02×10^6 to 7.19×10^6 ) yen for women, respectively.

Conclusions: The calculated additional LLE due to missed CRC screening was relatively small but suggests preventive health services should be considered in disaster response planning. These findings provide a quantitative framework for evaluating health impacts of service disruptions.

Comments

2. This will be dangerously lost (misled)on reader uncertain sbout colonoscopy in their practices (expert opinion public heslth actor needs to go to text)

Reply

Colonoscopy is generally recommended in all cases of positive feacal immunochemical test (FIT). The administration repeatedly contacts and recommends those who have not undergone it to undergo colonoscopy. In this estimation, the impact is estimated by using statistics from Japan on the number of colorectal cancers diagnosed from people who have undergone the FIT. The average colonoscopy compliance rate and the percentage of cancers detected in the process are factored in. The following has been added to the method regarding the usual response.

Page 9 Line 142-144

Colorectal cancer screening data of Minamisoma is compiled for those who participated in the annually recommended faecal immunochemical test (FIT). Colonoscopy is recommended if FIT is positive.

3.Please it needs lacks clear statements- tech jargon is not for abstract section

A)how many cancers were discovered at later staged updtsged (what stage 4,3,2 /10000 adults

Reply

The estimated missed cancers in relation to men and women are listed in the abstract as follows. Detailed estimates of the number of missed cancers per stage have been omitted from the abstract, as they are listed in table 3.

Abstract

Methods: We utilized the population and CRC screening data targeting 40-74 years-old residents in Minamisoma City. We compared the actual screening participation in 2011 with projected participation rates based on pre-disaster levels to estimate the number of residents who missed screening due to the disaster. Based on national CRC screening performance data and stage-specific survival rates in Japan, we estimated the number of missed CRC cases and modeled the additional the loss of life expectancy (LLE) due to CRC resulting from a one-year delay in diagnosis.

B)which population is target for extra intervention (young or elderly define age cut

Reply

We appreciate your comments. We have added the eli

Page 9 Line 139-140

We focused on individuals aged 40–74 years who were eligible for municipal CRC screening in Minamisoma in 2011.

C)How many otherwise avoidable deaths !!

Reply

The extent to which death occurs is not calculated in this simulation. In this case, life expectancy lost is calculated as an assessment of the risk caused by not undergoing screening. Although there are methods to calculate lost life expectancy from additional deaths that occur, we believe that these are not optimal when estimating the short-term impact in a small population such as in this case.

D)Is this enough evidence health loss , to support which/ sny salvage intervention ?

Whose (why do) criteria support a special intervention ?

Reply

We appreciate your important question. There are no clear criteria on whether intervention should be undertaken, and efforts should be made by disaster-affected areas to regain the effectiveness of their peacetime screening programmes compared to the priorities of other issues. As we describe in our discussion, the results of this simulation can be used for comparison with other health effects that have occurred since the disaster. From the perspective of lost life expectancy, we can conclude that the administrative priority should be on diabetes control, as the burden of diabetes is higher compared to the increase in diabetes that occurred after the disaster. On the other hand, from the CPLYS perspective, measures to increase participation in colorectal cancer screening are cost-effective, so it is reasonable to take measures to increase the declining participation rate. We should explore the desired intervention method, but it would be desirable to consider implementing a reintroduction of screening awareness and enhanced follow-up programmes to restore screening participation rates. We have added this issue as follows.

Page 31 Line 31-33

The desired intervention is a subject for future research, but it would be desirable to consider implementing a reintroduction of screening awareness and enhanced follow-up programmes to restore screening participation rates.

E) did they adjust for real - world routine non compliance

Reply

This is a important topic. In colorectal cancer screening, participation rates vary by region. The genelalizabitity in this regard is described in the limitaion below.

Page 38 Line 150-151

Finally, the Minamisoma City population was < 70,000 at the time of the disaster, which is a relatively small target population. This was compounded by the fact that the CRC screening participation rate was low (in the 10% range) before the disaster, suggesting a potentially limited number of study participants. These factors introduced significant limitations during the verification process. However, to mitigate these limitations, we employed Monte Carlo analysis to enhance the validity of our research.

Comments

If they are concerned that they underestimated the impsct this is a very concerning message ( this is a point for limitations not abstract )

Do They by extension dangerously downplay screening

( I don’t like that To msny doubt provokers come across my desk! History changing tech makes them wrong advice going forward

Reply

Our interpretation is also that the decline in colorectal cancer screening after the disaster should not be underestimated. We do not underestimate the impact of screening. We are only adding a note that the impact shown may be a numerical underestimation. In other words, the actual impact of the drop in screening uptake may be greater than that simulated here, and may require more vigilance. This is a Simulation study and there are usual limitations to the estimates.

---

## [Editor Report · Decision Letter 2]

2 May 2025

Estimating the Impact of Missed Colorectal Cancer Diagnoses on Life Expectancy in Minamisoma City Following the 2011 Triple Disaster

PONE-D-24-16149R2

Dear Dr. Saito,

We’re pleased to inform you that your manuscript has been judged scientifically suitable for publication and will be formally accepted for publication once it meets all outstanding technical requirements.

Kind regards,

Ahmed E. Abdel Moneim

Academic Editor

PLOS ONE
---

## [Editor Report · Acceptance letter]

PONE-D-24-16149R2

PLOS ONE

Dear Dr. Saito,

I'm pleased to inform you that your manuscript has been deemed suitable for publication in PLOS ONE. Congratulations! Your manuscript is now being handed over to our production team.

Kind regards,

on behalf of

Dr. Ahmed E. Abdel Moneim

Academic Editor

PLOS ONE